# Two invertible networks for the matrix element method

**Anja Butter[1], Theo Heimel[1], Till Martini[2★], Sascha Peitzsch[3★] and Tilman Plehn[1]**

**1** Institut für Theoretische Physik, Universität Heidelberg, Germany
**2** Fraunhofer Zentrum SIRIOS,
Fraunhofer Institute for High-Speed Dynamics EMI, Berlin, Germany
**3** Fraunhofer Zentrum SIRIOS, Fraunhofer Institute for Open Communication Systems
FOKUS, Berlin, Germany

## Abstract

The matrix element method is widely considered the ultimate LHC inference tool for small event numbers. We show how a combination of two conditional generative neural networks encodes the QCD radiation and detector effects without any simplifying assumptions, while keeping the computation of likelihoods for individual events numerically efficient. We illustrate our approach for the CP-violating phase of the top Yukawa coupling in associated Higgs and single-top production. Currently, the limiting factor for the precision of our approach is jet combinatorics.

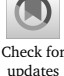

## Contents

---

★ Work on this article was conducted while employed at Humboldt-Universität zu Berlin, Institut für Physik, Berlin, Germany.

# 1    Introduction

In the search for optimal analysis methods at colliders, the matrix element method (MEM) has been playing a key role since it was developed for the Tevatron [1, 2]. If offers an especially simple and interpretable link between theory predictions and hypothesis tests. Its optimality is derived directly from the Neyman-Pearson theorem, which means it includes all available information encoded in phase space configurations and evaluates it using an optimal hypothesis test. The MEM is based on the observation that we can compute the likelihood of individual events, given a theory hypothesis, as the scattering amplitude from first-principle quantum field theory. Two different theory hypotheses or parameter points then define a likelihood ratio for a given event. The log-likelihood ratio of an event sample follows from adding individual events' log-likelihood ratios, but unlike essentially all other inference method the combination of a number of events into a distribution is not necessary.

The first application of the MEM was the Tevatron measurement of the top mass based on a limited number of statistically defined top quark events [3–6]. Also at the Tevatron, it was used to discover the single-top production process [7]. At the LHC, several studies [8–11] and applications [11–15] of the MEM exist. The challenge in applying the MEM is that we have to integrate over the scattering amplitudes at the parton level for each measured event. This makes MEM applications extremely CPU-expensive. For instance Madgraph can already compute parton-level amplitudes for a given event automatically at leading order [16]. To really use the power of the method we need to base it on precision predictions, including QCD jet radiation [17] and NLO QCD corrections, as shown for color-neutral particles in the final state [18, 19] and for jet production [20]. A consistent treatment of the MEM at NLO has been developed for electron-positron collisions with final-state radiation [21] and for hadronic collisions with modified jet algorithms [22], and standard jet algorithms [23, 24].

We will show how modern machine learning (ML) can enhance the MEM. Fast and invertible LHC simulations benefit from generative networks [25–27] like generative adversarial networks (GANs) [28], variational autoencoders (VAEs), normalizing flows, and their invertible network (INN) variant [29]. Within the established simulation chain, such networks can be applied to loop integrals [30], phase space integration [31, 32], phase space sampling [33–36], event subtraction [37], event unweighting [38, 39], parton showering [40–43], super-resolution enhancement [44, 45], or detector simulations [46–50]. Once we control the forward direction with NN-based event generators [51–56], conditional GANs and INNs also allow us to invert the simulation chain, to unfold detector effects [57–59] or to extract the hard scattering process in a statistically consistent manner [60, 61]. The fully calibrated inverted simulation uses the same conditional INN (cINN) as simulation-based inference [62, 63] or kinematic reconstruction [64]. Obviously, any application of (generative) networks to LHC physics requires an uncertainty treatment [56, 65]. A related ML-approach to likelihood extraction is based on simulated versus observed event samples [66]. A connection between the MEM and modern ML-methods, mentioned in Ref. [67], was demonstrated in Ref. [68], specifically using a deep regression network to evaluate the MEM integral.

In this paper we show how a combination of two cINNs allows for a better modeling of QCD and detector effects while keeping the MEM numerically efficient. First, a Transfer-cINN learns the effects of the parton shower, detector resolution, and reconstruction on simulated events. Second, an Unfolding-cINN provides a phase space mapping for the integration over the hard-scattering phase space at parton level. For both networks we use a Bayesian network version of the INN [56, 65] to track their reliability. The Bayesian Transfer-cINN also provides an uncertainty estimate for the extracted likelihood. Our toy example is the search for CP violation in the top Yukawa coupling, based on the kinematics of single top and Higgs production. In Sec. 2 we introduce the physics process and the effect of a CP-phase on the event kinematics.

Table 1: Cut flow for $pp \to tHj$ with $H \to \gamma\gamma$ and for SM events ($\alpha = 0°$). We assume $m_b = 0$ and intermediate on-shell particles.

| Dataset | cut | rate [ab] | fraction |
|---|---|---|---|
| leptonic | $\sigma$ | $43.6 \cdot 10^3$ | |
| | $\sigma \times \mathrm{BR}$ | 7.38 | |
| | $\geq 2$ photons with $p_T > 20$ GeV and $\eta < 2.4$ | 3.58 | 0.485 |
| | $\geq 1$ muon with $p_T > 20$ GeV and $\eta < 2.4$ | 2.29 | 0.310 |
| | $\geq 2$ jets | 1.69 | 0.230 |
| | 1 $b$-jet with $p_T > 25$ GeV and $\eta < 2.4$ | 1.00 | 0.136 |
| | $\geq 1$ jets with $p_T > 25$ GeV and $\eta < 2.4$ | 0.41 | 0.055 |
| hadronic, no ISR | $\sigma$ | $43.6 \cdot 10^3$ | |
| | $\sigma \times \mathrm{BR}$ | 44.28 | |
| | $\geq 2$ photons with $p_T > 20$ GeV and $\eta < 2.4$ | 19.56 | 0.442 |
| | $\geq 4$ jets | 7.09 | 0.160 |
| | 1 $b$-jet with $p_T > 25$ GeV and $\eta < 2.4$ | 3.93 | 0.089 |
| | $\geq 3$ jets with $p_T > 25$ GeV and $\eta < 2.4$ | 1.23 | 0.028 |
| hadronic, with ISR | $\sigma$ | $43.6 \cdot 10^3$ | |
| | $\sigma \times \mathrm{BR}$ | 44.26 | |
| | $\geq 2$ photons with $p_T > 20$ GeV and $\eta < 2.4$ | 18.37 | 0.415 |
| | $\geq 4$ jets | 12.67 | 0.286 |
| | 1 $b$-jet with $p_T > 25$ GeV and $\eta < 2.4$ | 6.44 | 0.146 |
| | $\geq 3$ jets with $p_T > 25$ GeV and $\eta < 2.4$ | 3.06 | 0.069 |

In Sec. 3 we introduce our dual-cINN architecture, which we benchmark on simulated events with a leptonic and hadronic top decay in Sec. 4. While we do not (yet) include a full NLO calculation of the event likelihood, we do combine different jet numbers through initial state radiation as a first step in this direction in Sec. 4.2. We discuss some remaining challenges for the network precision due to combinatorics once we include many jets. Once those issues can be overcome, our method naturally extends to NLO likelihood predictions.

## 2 LHC process

To illustrate how we can use generative networks for measurements using the MEM, we choose associated single-top and Higgs production

$$pp \to tHj. \tag{1}$$

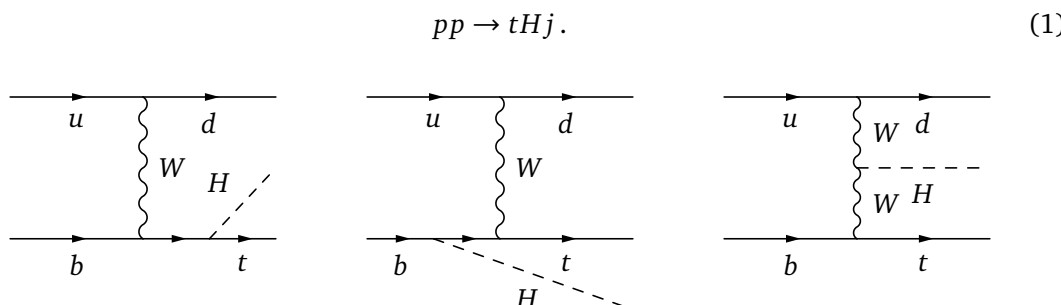

Figure 1: Leading-order Feynman diagrams for the hard process $pp \to tHj$. We neglect the second diagram in the limit of a massless bottom quark. The diagrams also appear with an inverted light-quark line.

Table 2: Fit parameters for the fiducial cross sections, for the formula given in Eq.(6).

| Dataset | $A$ [fb] | $B$ [fb] | $C$ [fb] | $D$ [fb] | $E$ [fb] |
|---|---|---|---|---|---|
| leptonic | $4.07 \cdot 10^{-4}$ | $2.37 \cdot 10^{-3}$ | $-1.22 \cdot 10^{-3}$ | $1.86 \cdot 10^{-6}$ | $-2.90 \cdot 10^{-7}$ |
| hadronic, no ISR | $1.23 \cdot 10^{-3}$ | $7.59 \cdot 10^{-3}$ | $-3.78 \cdot 10^{-3}$ | $1.24 \cdot 10^{-5}$ | $-7.96 \cdot 10^{-6}$ |
| hadronic, with ISR | $3.06 \cdot 10^{-3}$ | $2.05 \cdot 10^{-2}$ | $-9.50 \cdot 10^{-3}$ | $1.90 \cdot 10^{-5}$ | $-5.77 \cdot 10^{-6}$ |

This process will allow us to study a CP-phase in the top Yukawa coupling at future LHC runs [69–77], unfortunately with limited expected event numbers. This limitation means that we need an optimal analysis framework for this measurement, specifically the matrix element method based on likelihood ratios. We choose the decay $H \to \gamma\gamma$ to illustrate our point with a focus on the signal process. Our methods can be generalized to other decay processes, for which we would also need to include continuum backgrounds.

To extract the likelihoods corresponding to different theory hypotheses for a given phase space configuration, we will use a combination of two neural networks. The crucial ingredient for our NN training are paired events at the hard-scattering level and after parton shower and detector simulation. The usual Monte Carlo simulation starts from the hard scattering matrix element and successively adds parton showers and detector effects. For the simulation of our signal process we use Madgraph5, v3.1.0, with LO-NNPDF and $\alpha_s = 0.119$ [78]. We produce the heavy top and Higgs on their respective mass shells and decay them in a second step. Our simulation includes the standard Pythia [79] parton shower, Delphes [80] as a fast detector simulation, and Fastjet [81] to reconstruct anti-$k_T$ jets [82] of size 0.4. As illustrated in Fig. 1 we consider massless incoming bottoms, while the jet in the final state always comes from a light quark. In the Standard Model, the dominant contribution stems from the first diagram where the Higgs couples to the top. Throughout our analysis we neglect the second diagram because of the small bottom Yukawa.

We generate three different datasets. First, the $W$ decays leptonically, into a muon and a neutrino,

$$pp \to tHj \to (b\mu^+ \nu_\mu)(\gamma\gamma)j. \tag{2}$$

Second, the $W$ decays hadronically, resulting in two jets

$$pp \to tHj \to (bjj)(\gamma\gamma)j. \tag{3}$$

In both cases, we do not generate initial state radiation (ISR) and multi-parton interactions by disabling the corresponding settings in Pythia. For the third dataset, we again consider hadronic decays, but this time including ISR jets,

$$pp \to tHj \to (bjj)(\gamma\gamma)j + \text{QCD jets}. \tag{4}$$

We allow for up to four additional jets in our datasets. They can come from final state radiation, or, in the case of the third dataset, from initial state radiation. The total proton-proton cross section for $tHj$ production is 43.6 fb, where we always combine top and anti-top production. Table 1 provides an overview of the cross sections and the detector-level cuts. We do not apply cuts in $p_T$ or $\eta$ at the hard-scattering level. To illustrate the ML-based numerical approach to the MEM we limit ourselves to the more challenging signal, with the narrow Higgs mass peak, and ignore all continuum backgrounds.

**CP-violating Yukawa coupling**

To study the top Yukawa coupling independently of the top mass, we allow for a mixture of CP-even and CP-odd interactions [83]. The top-Higgs interaction is parameterized by

$$\mathcal{L}_{t\bar{t}H} = -\frac{y_t}{\sqrt{2}} \Big[ a \cos\alpha \, \bar{t}t + ib \sin\alpha \, \bar{t}\gamma_5 t \Big] H \,, \tag{5}$$

with $a = 1$ and $b = 2/3$ [84], so the total $gg \to H$ cross section remains constant when we vary $\alpha$. This model has only one free parameter, the CP-angle $\alpha$, interpolating between a CP-even ($\alpha = 0°$) and a CP-odd ($\alpha = 180°$) Yukawa coupling. Because of this coupling structure, all observables $O$ for the $tHj$ process obtained by integrating over the hard-scattering phase space take the functional form

$$O(\alpha) = A + B(1 - \cos\alpha) + \sin\alpha \, (C \sin\alpha + D + E \cos\alpha) \,, \tag{6}$$

as long as we only consider higher-order QCD corrections. A fit of the fiducial cross section to the angles $\alpha = -180°, -90°, -45°, 0°, 22.5°, 45°, 90°, 135°, 180°$ gives the parameters quoted in Tab. 2. With $D, E \ll A, B, C$ we see that there is an approximate degeneracy in the sign of the CP-phase.

In Fig. 2 we show the fiducial $tHj$ cross section including decays after cuts as a function of $\alpha$. Typical rates especially around the Standard Model ($\alpha = 0°$) are below 0.01 fb, which means that in an actual analysis we need to extract as much information as possible from a small number of events and their kinematic features. The rate increase with $\alpha$ is driven by the interplay of the leading top Yukawa contribution, shown to the left in Fig. 1, and the subleading gauge coupling to the right. The angle $\alpha$ defines a relative phase between the two diagrams, which leads to a destructive interference in the Standard Model. The change in the total rate reflects the shift from destructive to constructive interference. From Fig. 2 we expect that small values $\alpha \lesssim 40°$ will hardly be measured from the total rate, especially once we include experimental and theoretical uncertainties. This means we have to complement the rate information with kinematic features.

We show the distributions for the hard-scattering $tHj$ kinematics in Fig. 3. Again, the change in the kinematics is driven by the interference between the leading top Yukawa contribution and the subleading gauge contribution. For $p_{T,t}$ and $p_{T,H}$, large phases lead to a harder transverse recoil of the heavy particles and a less central scattering process in rapidity. In contrast, in the Standard Model the two leading Feynman diagrams cancel in the central region. In the angular separation $\Delta R_{tj}$ a second maximum with a large angular separation vanishes when we switch from destructive to constructive interference. Unlike for the total rate we see that changing $\alpha$ from zero to 45° leads to visible kinematic shifts.

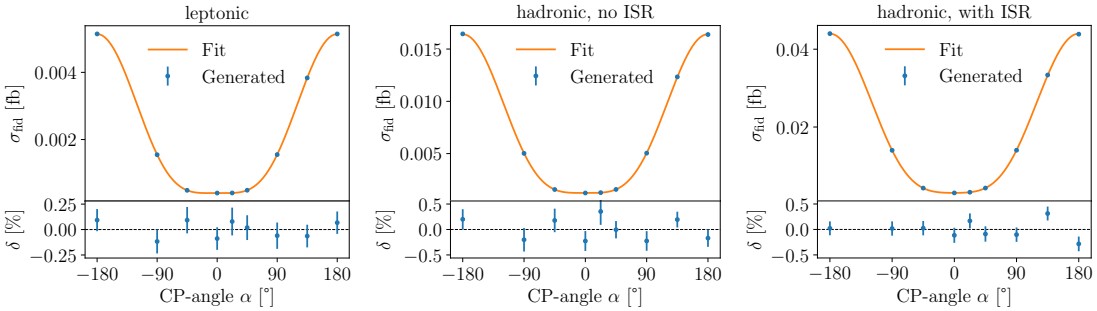

Figure 2: Fiducial cross section including decays and after cuts as a function of the CP-angle $\alpha$. The lower panels illustrate the agreement between the generated data and the fitted continuous function defined in Tab. 2.

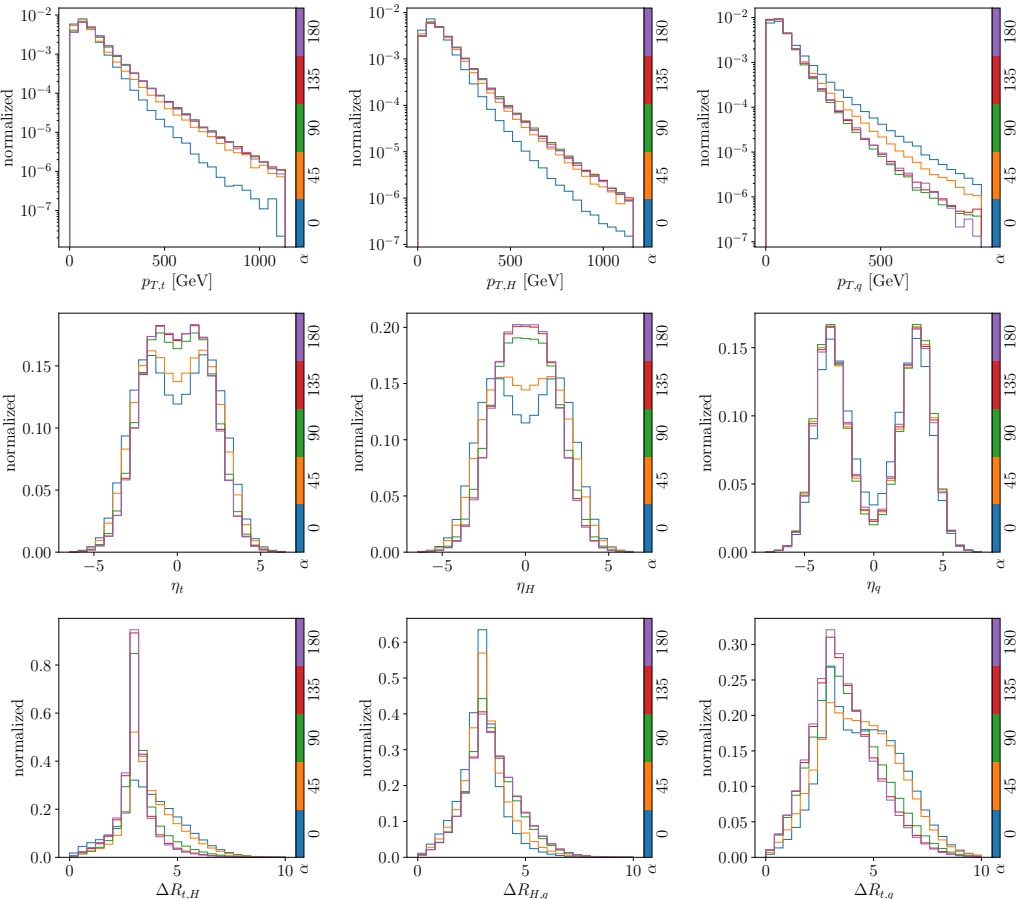

Figure 3: Kinematic distributions for the hard-scattering $tHj$ final state for different values of the CP-angle.

**Data samples**

As mentioned above, we will work with three different datasets, consisting of paired events at the hard-scattering level and after parton shower, detector simulation, and reconstruction. All samples share the same format for the hard scattering, including the CP-angle $\alpha$ and the 4-momenta $\{p_t, p_H, p_q\}$. For the leptonic top decay, Eq.(2), the reco-level events are described by the 4-momenta $\{p_{\gamma,1}, p_{\gamma,2}, p_b, p_\mu, p_{j,1}\}$. While we could use established methods to reconstruct the neutrino momentum from the missing transverse energy, we do not include it as an additional input to the network. Since it is a deterministic function of the other momenta, it would not increase the number of degrees of freedom and instead, only make the training more challenging. Additional light jets can come from final state radiation, and the photon and jet momenta are ordered in $p_T$. We do not allow for initial state radiation, to simplify the problem as much as possible. We train our networks on 1.3M paired events with $\alpha$ drawn from a uniform distribution. The test dataset consists of 260k events for each of the angles $\alpha \in \{0°, 45°, 90°, 135°, 180°\}$.

The other two datasets assume a hadronic top decay, without ISR (Eq.(3)) and with ISR (Eq.(4)). The reco-level event format includes $\{p_{\gamma,1}, p_{\gamma,2}, p_b, p_{j,1}, p_{j,2}, p_{j,3}\}$, plus potential additional light jet momenta. For the hadronic decays, the network has to learn which of the jets comes from the hard scattering. This problem gets more challenging when we include ISR, because the additional jets can lead to faulty or incomplete reconstruction. Hence, we lose the clear correspondence between parton-level and reconstructed jets in the high-multiplicity

events. We extract the hard-scattering momenta before we add ISR in our simulation chain, so our method does not require a boost into the hard-scattering rest frame. In the ISR case, we increase the number of training events to 3.4M.

## 3  ML-MEM

For our signal-only toy example, the matrix element method tracks the dependence of the hard scattering cross section on one continuous parameter of interest, the CP-phase $\alpha$ appearing in the Lagrangian of Eq.(5). We denote the hard-scattering momenta at parton level as $x_{\text{hard}}$ and split the differential cross section into a total cross section and a probability density,

$$\frac{\mathrm{d}\sigma(\alpha)}{\mathrm{d}x_{\text{hard}}} = \sigma(\alpha)\, p(x_{\text{hard}}|\alpha)\,. \tag{7}$$

The likelihood for a single hard-scattering event $x_{\text{hard}}$ to correspond to a given value for $\alpha$ is then

$$p(x_{\text{hard}}|\alpha) = \frac{1}{\sigma(\alpha)}\, \frac{\mathrm{d}\sigma(\alpha)}{\mathrm{d}x_{\text{hard}}}\,. \tag{8}$$

Next, we consider the effects of parton shower, hadronization, detector, and reconstruction. The corresponding forward simulation maps $x_{\text{hard}}$ to a reco-level configuration $x_{\text{reco}}$, provided it passes the cuts. In the forward direction $p(x_{\text{reco}}|x_{\text{hard}})$ is the conditional probability for a reco-level event $x_{\text{reco}}$, given $x_{\text{hard}}$ at parton level. In general, this conditional probability depends on our parameter of interest, $p(x_{\text{reco}}|x_{\text{hard}},\alpha)$, so we can use it to write the likelihood linking a single reco-level event $x_{\text{reco}}$ to the parameter $\alpha$ as

$$
\begin{aligned}
p(x_{\text{reco}}|\alpha) &= \int \mathrm{d}x_{\text{hard}}\, p(x_{\text{hard}}|\alpha)\, p(x_{\text{reco}}|x_{\text{hard}},\alpha) \\
&= \frac{1}{\sigma(\alpha)} \int \mathrm{d}x_{\text{hard}}\, \frac{\mathrm{d}\sigma(\alpha)}{\mathrm{d}x_{\text{hard}}}\, p(x_{\text{reco}}|x_{\text{hard}},\alpha)\,.
\end{aligned} \tag{9}
$$

The conditional probability $p(x_{\text{reco}}|x_{\text{hard}},\alpha)$ corresponds to the usual transfer function, which can sometimes be approximated by a Gaussian. In general, it is only defined implicitly through a complex forward simulation. Using the single-event likelihoods at the reco-level we can compute the likelihood for an event sample as a function of the parameter of interest,

$$
\begin{aligned}
L(\alpha) &\approx \prod_{\text{events } j} p(x_{\text{reco},j}|\alpha) \\
&= \prod_{\text{events } j} \frac{1}{\sigma(\alpha)} \int \mathrm{d}x_{\text{hard}}\, \frac{\mathrm{d}\sigma(\alpha)}{\mathrm{d}x_{\text{hard}}}\, p(x_{\text{reco},j}|x_{\text{hard}},\alpha)\,,
\end{aligned} \tag{10}
$$

where we omit any prefactors related to the observed number of events [27].

**Transfer-cINN**

Because of the phase space cuts, the conditional probability in Eq.(9) is not normalized to one. Instead, we can define the acceptance rate $a(x_{\text{hard}},\alpha)$ to obtain

$$\int \mathrm{d}x_{\text{reco}}\, p(x_{\text{reco}}|x_{\text{hard}},\alpha) = a(x_{\text{hard}},\alpha)\,. \tag{11}$$

Alternatively, we can account for this efficiency by replacing the full volume with the fiducial volume at the hard-scattering level. Here we assume that there is a hard cut-off at the parton level $x_{\text{hard}}$, even though we define our cuts at the detector-level. This means we replace $a(x_{\text{hard}}, \alpha) \neq 1$ by shifting $\sigma(\alpha) \rightarrow \sigma_{\text{fid}}(\alpha)$ in Eq.(9),

$$
\begin{aligned}
p(x_{\text{reco}}|\alpha) &= \int_{\text{fid}} \mathrm{d}x_{\text{hard}} \, p(x_{\text{hard}}|\alpha) \, p(x_{\text{reco}}|x_{\text{hard}}, \alpha) \\
&= \frac{1}{\sigma_{\text{fid}}(\alpha)} \int_{\text{fid}} \mathrm{d}x_{\text{hard}} \, \frac{\mathrm{d}\sigma(\alpha)}{\mathrm{d}x_{\text{hard}}} \, p(x_{\text{reco}}|x_{\text{hard}}, \alpha) \,.
\end{aligned}
\tag{12}
$$

In this integral, the differential cross section is available numerically and $p(x_{\text{reco}}|x_{\text{hard}}, \alpha)$ can be encoded in a neural network. Normally, this would be a regression task, but in our case we do not have the explicit training data to train a regression network. Instead, we train a normalizing flow, specifically a conditional cINN, to reproduce the reco-level kinematics for a given hard-scattering event from Gaussian random numbers

$$
r \sim p(r) \quad \longleftrightarrow \quad x_{\text{reco}} \sim p(x_{\text{reco}}|x_{\text{hard}}, \alpha) \qquad \text{Transfer-cINN.} \tag{13}
$$

In the inverse direction, this network estimates and encodes the conditional density over the reco-level phase space, and in the forward direction, it is nothing but a fast detector simulation generating reco-level events. We will see that for our purpose and implementation we can ignore the $\alpha$-dependence of $p(x_{\text{reco}}|x_{\text{hard}}, \alpha)$. The best way to train the network is to use data with variable $\alpha$ and ignore this input. Such a training improves the phase space coverage even for extreme values of $\alpha$ and provides an averaging over any remaining $\alpha$-dependence.

**Unfolding-cINN**

Even with a fast surrogate integrand, the integral in Eq.(12) is numerically challenging, because the squared matrix element spans several orders of magnitude and $p(x_{\text{reco}}|x_{\text{hard}}, \alpha)$ drops rapidly away from the trivial mapping of the intermediate on-shell particles and hard partons turning into single jets. We can define an appropriate mapping $x_{\text{hard}} \rightarrow q(x_{\text{hard}})$, or sampling

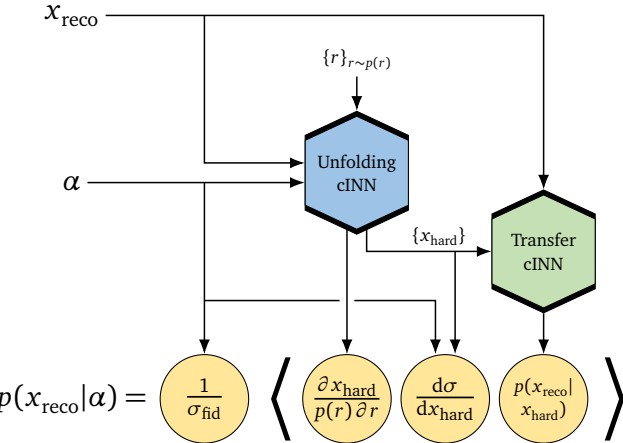

Figure 4: Dual-cINN setup of the MEM integrator evaluating Eq.(16) through sampling $r$. The Unfolding-cINN is conditioned on the CP-angle $\alpha$ and the reco-level event $x_{\text{reco}}$. The Transfer-cINN is conditioned on the hard-scattering event $x_{\text{hard}}$.

of $x_{\text{hard}} \sim q(x_{\text{hard}})$, such that Eq.(12) becomes

$$
\begin{aligned}
p(x_{\text{reco}}|\alpha) &= \int_{\text{fid}} dx_{\text{hard}}\, p(x_{\text{hard}}|\alpha)\, p(x_{\text{reco}}|x_{\text{hard}}, \alpha) \\
&= \left\langle \frac{1}{q(x_{\text{hard}})}\, p(x_{\text{hard}}|\alpha)\, p(x_{\text{reco}}|x_{\text{hard}}, \alpha) \right\rangle_{x_{\text{hard}} \sim q(x_{\text{hard}})} \\
&= \left\langle p(x_{\text{reco}}|\alpha) \right\rangle_{x_{\text{hard}} \sim q(x_{\text{hard}})} \qquad \Longleftrightarrow \qquad x_{\text{hard}} \sim q(x_{\text{hard}}) = p(x_{\text{hard}}|x_{\text{reco}}, \alpha).
\end{aligned}
\tag{14}
$$

The last line uses Bayes' theorem and means our $x_{\text{hard}}$-integration would become trivial if we could sample $x_{\text{hard}}$ from the distribution $p(x_{\text{hard}}|x_{\text{reco}}, \alpha)$.

To sample the hard-scattering momenta $x_{\text{hard}}$ following such a distribution we again train a conditional INN, mapping random numbers with a latent distribution $p(r)$ to the target distribution in momentum space,

$$
r \sim p(r) \quad \longleftrightarrow \quad x_{\text{hard}}(r) \sim p(x_{\text{hard}}|x_{\text{reco}}, \alpha) \qquad \text{Unfolding-cINN.}
\tag{15}
$$

It turns out that sampling $x_{\text{hard}}$ from $p(x_{\text{hard}}|x_{\text{reco}}, \alpha)$ defines the standard cINN used for unfolding or Bayesian inference in Refs. [60–62]. A better modeling of the distribution of $x_{\text{hard}}$ will lead to a more efficient integration. In Eq.(12) the Unfolding-cINN transforms the $x_{\text{hard}}$-integration into an $r$-integration. In the corresponding Jacobian we have to account for the full conditional dependence of $x_{\text{hard}}(r; x_{\text{reco}}, \alpha)$,

$$
\begin{aligned}
p(x_{\text{reco}}|\alpha) &= \frac{1}{\sigma_{\text{fid}}(\alpha)} \int dr\, \frac{\partial x_{\text{hard}}(r; x_{\text{reco}}, \alpha)}{\partial r} \left[ \frac{d\sigma(\alpha)}{dx_{\text{hard}}} p(x_{\text{reco}}|x_{\text{hard}}, \alpha) \right]_{x_{\text{hard}}(r; x_{\text{reco}}, \alpha)} \\
&\equiv \frac{1}{\sigma_{\text{fid}}(\alpha)} \left\langle \frac{1}{p(r)} \frac{\partial x_{\text{hard}}(r; x_{\text{reco}}, \alpha)}{\partial r} \left[ \frac{d\sigma(\alpha)}{dx_{\text{hard}}} p(x_{\text{reco}}|x_{\text{hard}}, \alpha) \right]_{x_{\text{hard}}(r; x_{\text{reco}}, \alpha)} \right\rangle_{r \sim p(r)}.
\end{aligned}
\tag{16}
$$

The dual-network architecture of our MEM integrator is illustrated in Fig. 4.

**Network architecture**

Both cINNs are built as a sequence of rational quadratic spline coupling blocks [85], each followed by a random rotation matrix. The spline coupling blocks implement a mapping between hypercubes. To make them compatible with a Gaussian latent space and the rotation matrices, we set their bounds to $[-10, 10]$. This ensures that the points passed through the network are sufficiently far from the spline boundaries, after applying a standard scaling to the training data. For a cINN that maps a batch of $B$ data points $x_i$ to points $r_i$ in a Gaussian latent space, given a condition $c_i$, the loss function is [27, 60]

$$
\mathcal{L}_{\text{cINN}} = \sum_{i=1}^{B} \left( \frac{r_i(x_i; c_i)^2}{2} - \log \left| \frac{\partial r_i(x_i; c_i)}{\partial x_i} \right| \right).
\tag{17}
$$

For the two networks we identify

$$
\begin{aligned}
x &= x_{\text{hard}}, & c &= (x_{\text{reco}}, \alpha) & &\text{Unfolding-cINN,} \\
x &= x_{\text{reco}}, & c &= x_{\text{hard}} & &\text{Transfer-cINN.}
\end{aligned}
$$

The networks are implemented in PyTorch [86]. We use the Adam [87] optimizer with a one-cycle learning rate scheduling [88]. After tuning the hyper-parameters of the Unfolding-cINN, we found that the same setup and hyper-parameters also yield good results for the Transfer-cINN. The network hyper-parameters are given in Tab. 3.

**Uncertainty-aware cINN**

Bayesian neural networks allow us to test the training stability and to estimate uncertainties on the network output. They take the architecture of standard regression, classification, or generative networks and allow the individual network weights to fluctuate. The uncertainty on the network output is then estimated by sampling from the weight distributions [89–94]. For generative networks this concept has been applied to normalizing flows or INNs [56, 65]. Here the network encodes a density over phase space and the uncertainty on this density over the same phase space. For more details on the Bayesian cINN we refer to the original papers [56, 65] and the lecture notes of Ref. [27]. By construction, Bayesian networks include an L2-regularization, so with limited extra numerical effort Bayesian networks deliver the same level of performance as their deterministic counterparts.

Because the Unfolding-cINN is only used to improve the importance sampling for the numerical integration, its uncertainty is irrelevant for the actual integral, so we do not generalize it to a Bayesian version for our final application. However, we will use a Bayesian Unfolding-cINN to estimate its uncertainties and confirm its reliability.

In contrast, the Transfer-cINN encodes the reco-level phase space density, which means we can use a Bayesian cINN to estimate the uncertainty of this learned density. Whenever we show results for this density, we also indicate the corresponding uncertainty from the network training. For tricky applications like the MEM this additional check on the network performance is crucial.

## 4 Performance

To illustrate the advantages and the remaining challenges of a ML-realization of the matrix element method, we show results for the associated Higgs and single-top production. We only consider signal events, because of the especially challenging Higgs mass pole. We test the two cINNs independently, including an uncertainty analysis through a Bayesian network setup.

### 4.1 Leptonic top decay

The first results we discuss in detail are for the leptonic top decay, as defined in Eq.(2). We start with a test of the Transfer-cINN from Eq.(13) in the forward direction. As mentioned above, the network generates 4-momenta of five final state particles at reco-level, including one light jet.

Table 3: Identical setup and hyper-parameters for the Transfer-cINN and the Unfolding-cINN.

| Parameter | cINN |
|---|---|
| Blocks | 20 |
| Block type | Rational quadratic |
| Layers per block | 5 |
| Units per layer | 256 |
| Spline bins | 30 |
| Epochs (Bayesian) | 100 (200) |
| Learning rate scheduling | One-cycle |
| Initial learning rate | $1 \cdot 10^{-4}$ |
| Maximum learning rate | $3 \cdot 10^{-4}$ |
| Batch size | 1024 |
| Training events | 1.3M |

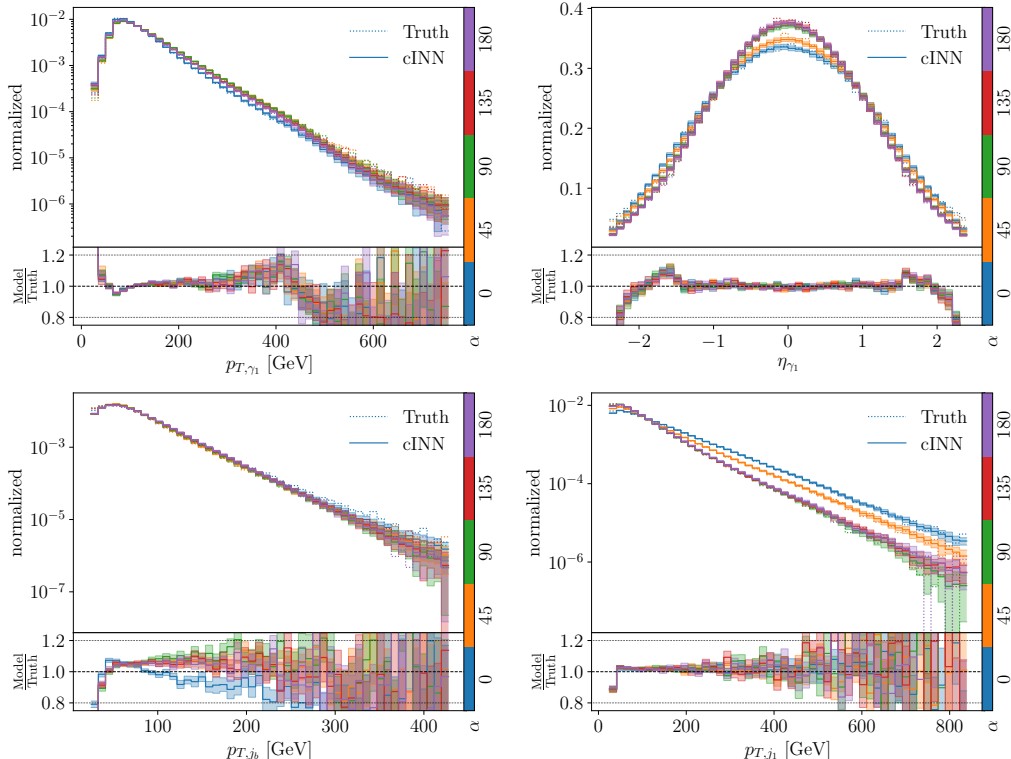

Figure 5: Forward-simulated kinematic distributions for the leptonic top decay, assuming five different CP-angles and including uncertainties from the Bayesian cINN. These distributions test the Transfer-cINN.

Assuming these particles to be produced on their mass shell we remove the photon and muon energies from the network's degrees of freedom, leaving us with a phase space dimensionality of $5 \cdot 4 - 3 = 17$. The forward generation is conditioned on the corresponding set of three hard-scattering momenta, all of them assumed to be on-shell. In all cases we implement a standard scaling, including a subtraction of the mean values and a normalization to standard deviation one. All hyper-parameters are shown in Tab. 3.

The kinematic distributions from the network evaluated in the forward direction are shown in Fig. 5, compared to the true reco-level distributions from the training dataset. One assumption we can test is that the Transfer-cINN does not have to be conditioned on $\alpha$, which means that this detail of the underlying model is numerically irrelevant. To make such a statement we need the uncertainties of the network prediction as a reference measure. The reliability of the network is best seen in the lower panels, where we see that deviations from the true distributions appear in the tails of the distributions. The uncertainty estimate is reliable in the bulk of all distributions, reflects the increased uncertainty in the $p_T$-tails, and covers the rapidly dropping rapidity distributions less well. Looking at this uncertainty we confirm that the distributions differ for varying $\alpha$, but this variation is explained entirely by the effects on the hard-scattering distributions, combined with an $\alpha$-independent Transfer-cINN.

Next, we test the Unfolding-cINN, which we will use to improve the numerical integration. The three generated momenta are defined at the parton level, all particles are on-shell, and we assume momentum conservation in the azimuthal plane. The corresponding 7-dimensional phase space is spanned by the coordinates $(\vec{p}_t, \vec{p}_h, p_q^z)$. The conditional input is the reco-level phase space, where we allow for up to four additional jets, *i.e.* altogether up to nine 4-momenta zero-padded. In addition, we condition on the angle $\alpha$. As for the Transfer-cINN we implement a standard scaling for all data. The results for the unfolding, with uncertainties, are shown in

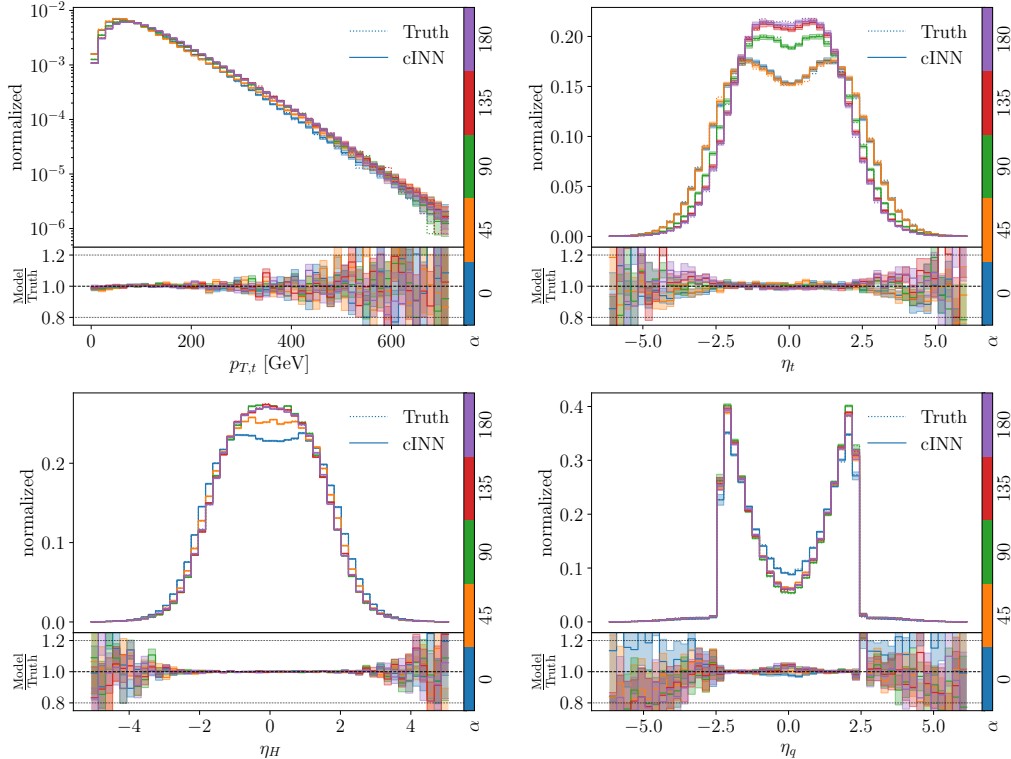

Figure 6: Unfolded kinematic distributions for the leptonic top decay, assuming five different CP-angles and including uncertainties from the Bayesian cINN. These distributions test the Unfolding-cINN.

Fig. 6. Again, we see that the network reproduces all features, including the $\alpha$-dependence, and remaining differences between the cINN-unfolded and truth events are covered by the network uncertainties.

After testing both networks individually, we can use their combination to extract likelihoods as a function of the CP-angle $\alpha$ for a given set of reco-level events. While our method allows us to compute these likelihood for individual events, we only show combinations of 400 events, to see if the corresponding distributions are reliable. In the center panels of Fig. 7 we show likelihood distributions for an assumed true value $\alpha = 45°$. According to Fig. 3 we expect the event kinematics to be comparably sensitive to CP-angles around this value.

We compute the negative log-likelihood $-2 \log L_i(\alpha)$ for a given event $i$ from the integral given in Eq. 16, evaluated for 100k sampling points. To improve the numerical stability we use trimmed means and standard deviations for the integration, which means we leave out 1% of random numbers in the lower and upper tails when computing the integral. Also in the integration we remove rare unphysical configurations, for instance when the unfolding network generates events with momentum fractions $x > 1$ or the leading-order differential cross section turns negative. The log-likelihoods for individual events is then added to give smooth log-likelihood distributions for small event samples. In the upper central panel of Fig. 7 we show the likelihood values for 400 events as a function of $\alpha$. We show the actual data points as well as a polynomial fit to those points. The uncertainties on the log-likelihood are computed using Gaussian error propagation of the Monte Carlo integration error. Because we are interested in likelihood ratios, we always show the difference in the log-likelihoods to the minimum of the fitted curve. In the lower panel we show the results from the Bayesian Transfer-cINN, where we visualize the training uncertainty by repeating the likelihood calculation for 10 networks sampled from the distribution over their trainable weights. Comparing the two panels

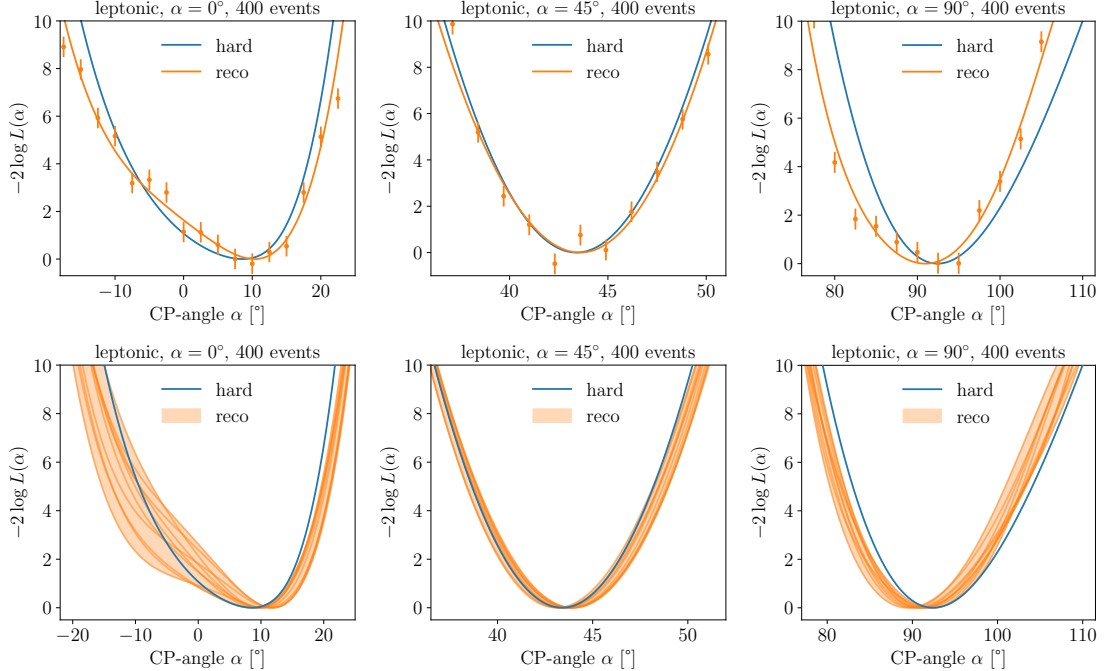

Figure 7: Likelihoods for the leptonic top decay as a function of the CP-angle $\alpha$, extracted from 400 events for three assumed truth angles. For the Bayesian uncertainties we show the integrated likelihoods from 10 sampled networks.

we see that the variation of points around the polynomial fit are what we expect from the network uncertainties. The deviation from the hard-scattering truth distribution shows a small, insignificant shift, which might come from the reconstruction of the longitudinal neutrino momentum.

In the outer panels of Fig. 7 we show the same results for assumed CP-angles of $0°$ and $90°$. The general pattern is the same as for $45°$, but we see that the quality of the measurement decreases for larger angles and becomes a challenge towards the SM-value. The reason can be seen in Fig. 3, namely that the effect of small shifts in the angle on the event kinematics is smaller than for $\alpha = 45°$. An additional complication for the SM-value $\alpha = 0°$ is that the total rate is symmetric under a sign flip of the CP-angle, and Eq.(6) shows that this symmetry is approximately also true for the kinematic distributions.

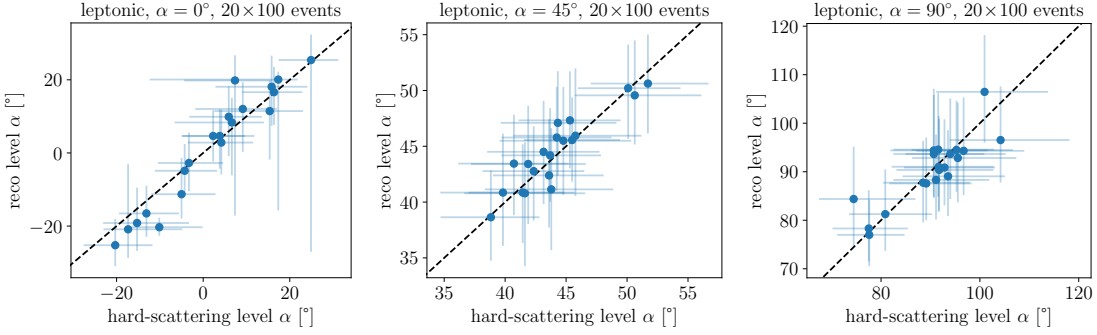

Figure 8: Calibration of the $\alpha$-measurement from leptonic top decays, in terms of mean values and 68% confidence intervals extracted from 20 sets of 100 events at parton level and measured.

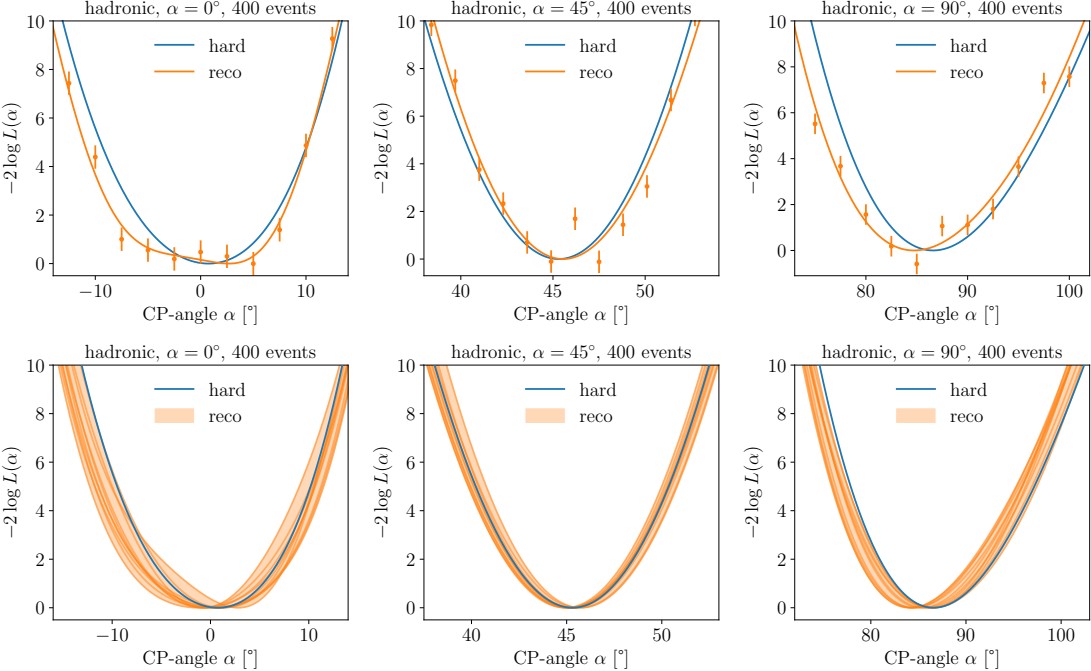

Figure 9: Likelihoods for the hadronic top decay as a function of the CP-angle $\alpha$, extracted from 400 events for three assumed truth angles. For the Bayesian uncertainties we show the integrated likelihoods from 10 sampled networks.

Finally, we check the calibration of the extracted CP-angle for 20 sets of 100 events defined at different angles. For each of those sets we extract a mean and a 68% confidence interval on the hard-scattering level truth and extracted angles. We compute the confidence intervals by assuming an approximately Gaussian likelihood distribution with different lower and upper tails, such that the likelihood values at the two limits are the same. This can lead to asymmetric error bars. Instead of testing our method on a large number of different CP phases, we use a small number of values for $\alpha$ and show the correlation between the MEM result at the reco- and hard-scattering level for several sets of events to obtain approximate calibration curves. In Fig. 8 we show such curves for the three assumed true $\alpha$-values. For the best measurement at $\alpha = 45°$ the true and extracted values of the angle are nicely correlated. A slight bias towards overestimating the angle can be removed through a proper calibration. For $\alpha = 90°$ the situation is similar, but, if anything, the bias tends to underestimate the true angle. Finally, for the challenging SM-value $\alpha = 0°$ the range of the correlation and the error bars increase, but the calibration is perfectly fine.

## 4.2 Hadronic top decay

Moving on to the more challenging hadronic top decay, we use the same neural network setup as before and see how it deals with the challenge moving from the neutrino reconstruction to increasingly complex jet combinatorics.

**Without ISR**

The only change between the leptonic top decay study and the hadronic top decays is that the reco-level phase space now covers two on-shell photons, one $b$-jet and three light-flavor jets, leading to $6 \cdot 4 - 2$ dimensions. In Fig. 9 we show the extracted likelihood distributions for 400 events, to be compared with Fig. 7 for the leptonic case. We see that the results are

completely comparable, which means that the additional complication of having to separate *W*-decay jets from the forward jet is not a problem for the networks. In Fig. 10 we see a new feature, as compared to Fig. 8, where for the SM-value $\alpha = 0°$ the networks sometimes chooses a mismatch of the sign of the angle between the hard-scattering level and the reconstruction level. This reflects the approximate symmetry from Eq.(6) and does not affect the likelihood extraction in a significant way.

**With ISR**

The situation changes when we allow for initial state radiation (ISR). In the absence of jets radiated from the initial state the network only has to distinguish decay jets from the forward jet in the hard process. The only difference between the two analyses without ISR and this one is that we now use a larger training dataset with 3.4M events, so the network can learn the more complex kinematic patterns. From Fig. 3 we know that the kinematic distribution of this hard forward jet, relative to the top and Higgs, is intimately tied to the CP-angle $\alpha$. Final state radiation can lead to a third decay jet or a splitting of the hard forward jet, in both cases not affecting the event topology much. This is different for ISR, because the additional jets can look similar to the hard forward jet, but they are really not part of the hard process and therefore only indirectly sensitive to the CP-angle. This makes it much more complicated to evaluate the hard-scattering likelihood. In general, the hadronic top decay combined with ISR breaks the one-to-one correspondence of hard-scattering partons and jets, which we have confirmed in detail, for instance using geometric correlations.

In the upper panels of Fig. 11 we show the unfolded kinematic distributions for the top, Higgs, and forward jet from the hard process. The Unfolding-cINN generally does well in reconstructing the hard process, which means the phase space integration as part of the MEM remains efficient after we include ISR.

In the lower panels of Fig. 11 we test the Transfer-cINN. We omit the kinematic distributions related to the top and *W*-kinematics, where the network does essentially as well as without ISR, and only show the critical distributions related to the jets. Here we can see that the performance of the Transfer-cINN is significantly worse, with typical deviations of up to 20% on the underlying phase space density. At this level the Bayesian uncertainty does not cover the difference between the truth and the cINN-generated distributions, and the size of the deviations is going to affect the extraction of the CP-angle. These results can immediately be generalized to the forward simulation of QCD jet radiation and detector effects.

As before, we show the extracted likelihood as a function of the CP-angle $\alpha$ in Fig. 12. For the deterministic Transfer-cINN we find that the likelihoods extracted from sets of 400 events

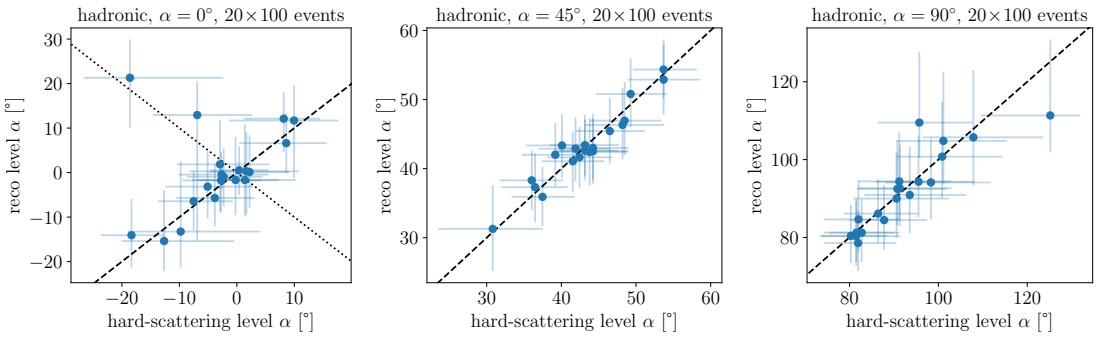

Figure 10: Calibration of the $\alpha$-measurement from hadronic top decays, in terms of mean values and 68% confidence intervals extracted from 20 sets of 100 events at hard-scattering level and measured.

reproduce the idealized hard-scattering level results fairly well. Problems occur around the SM-value, where we know that the effects of $\alpha$ only grow slowly. We find that for the shown event sample values $\alpha \lesssim 10°$ cannot really be distinguished from the Standard Model. This impression is confirmed by the uncertainties from the Bayesian network, indicating that there is a significant loss in sensitivity compared to the idealized hard-scattering level. On the positive side, the insensitive range of $\alpha \lesssim 10°$ is to be compared to the insensitive range of $\alpha \lesssim 45°$ for the total rate, as seen in Fig. 2. For the largest assumed angle $\alpha = 90°$ there is also a bias towards an underestimation of $\alpha$, which we have confirmed to be a general feature.

We find the same issues in the calibration curves shown in Fig. 13. Around $|\alpha| \lesssim 10°$ the network finds hardly any sensitivity to the mixing angle. The situation improves for the most sensitive region around $\alpha = 45°$, and for $\alpha = 90°$ the measurement indicates a bias which can, however, be removed through a standard calibration of the $\alpha$-measurement.

## 5 Outlook

The matrix element method is the method of choice to measure fundamental Lagrangian parameters from a small number of events at colliders. At hadron colliders, this method is computationally challenging. We have presented a way to compute the likelihoods at the hard-scattering level for reconstruction-level events with the help of two conditional generative

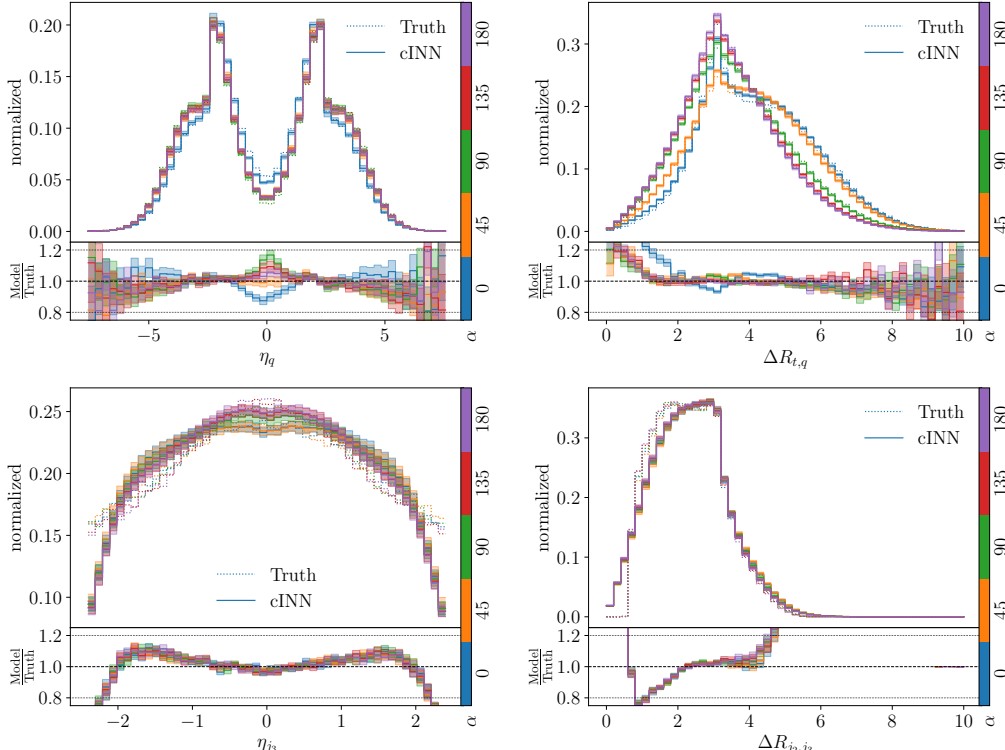

Figure 11: Top: unfolded kinematic distributions of the forward quark for hadronic top decay with ISR, assuming five different CP-angles and including uncertainties from the Bayesian cINN. These distributions test the Unfolding-cINN. Bottom: forward-simulated kinematic distributions for the hadronic top decay with ISR, assuming five different CP-angles and including uncertainties from the Bayesian cINN. These distributions test the Transfer-cINN.

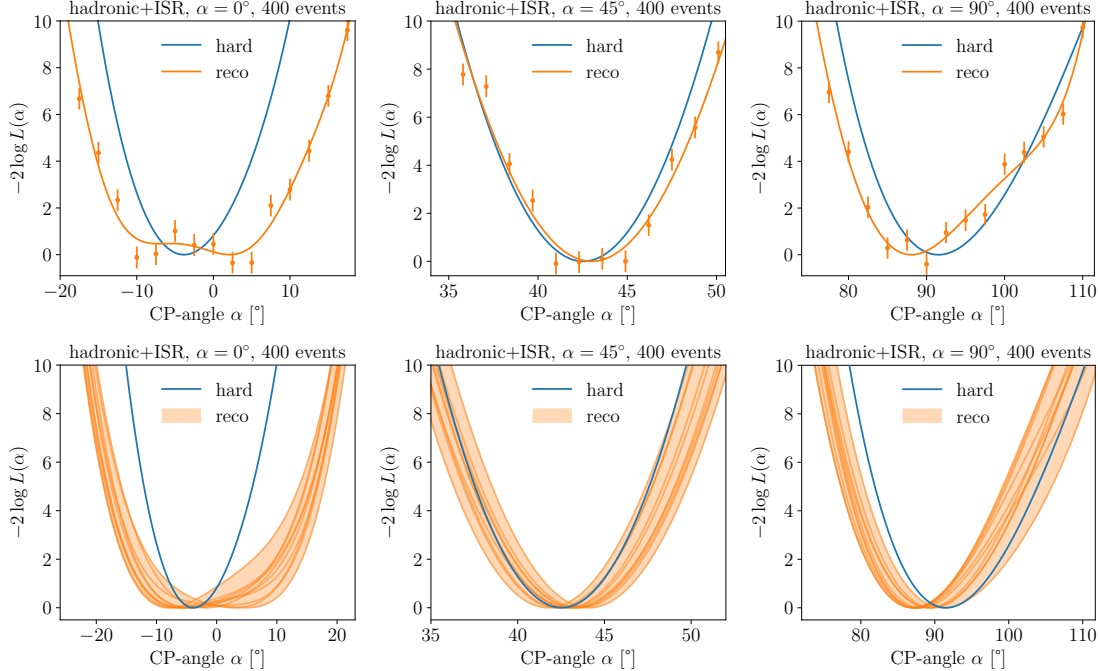

Figure 12: Likelihoods for the hadronic top decay, including ISR, as a function of the CP-angle $\alpha$, extracted from 400 events for three assumed truth angles. For the Bayesian uncertainties we show the integrated likelihoods from 10 sampled networks.

neural networks, specifically two cINNs. First, a Transfer-cINN encodes the effects of QCD jet radiation and detectors in a forward simulation. This network is nothing but a fast detector simulation, conditioned on the hard process. Second, the established Unfolding-cINN allows us to efficiently compute the integration over the hard-scattering phase space, just like a learned phase space mapping. In combination, the two conditional generative networks allow us to compute event-wise likelihood ratios efficently and without any assumptions on the form of any transfer function.

We illustrate our method using the extraction of the CP-angle in the top Yukawa coupling, accessible at the LHC through the associated production of a Higgs with a single top quark. Around the SM-value $\alpha = 0°$ the total rate of this process shows essentially no dependence on the CP-angle. We show how this changes once we include the full kinematic information through the MEM. For a leptonic top decay and for the hadronic top decay without initial state radiation our method shows a sensitivity close to the truth at the hard-scattering level. Once we include ISR, the combinatorics become more challenging, because the correspondence of jets and partons is broken and additional jets are hard to distinguish from the forward jets of the hard process. Nevertheless, even the results with ISR are promising and indicate that the MEM will allow us to extract maximal information for LHC processes with a small number of expected events.

## Acknowledgments

We would like to thank Peter Uwer for his continuing support and advice.

**Funding information** TP and AB would like to thank the Baden-Württemberg-Stiftung for support through the program *Internationale Spitzenforschung,* project *Uncertainties — Teach-*



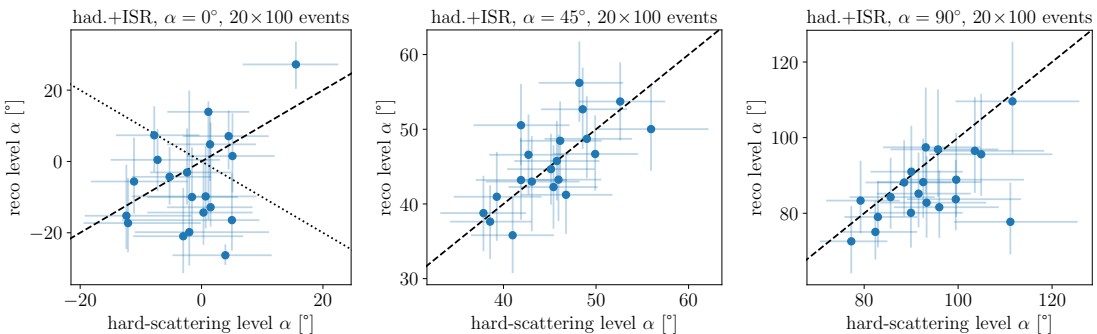

Figure 13: Calibration of the $\alpha$-measurement from hadronic top decays with ISR, in terms of mean values and 68% confidence intervals extracted from 20 sets of 100 events at hard-scattering level and measured.

ing AI its Limits (BWST_IF2020-010). AB and TP are supported by the DFG under grant 396021762 – *TRR 257 Particle Physics Phenomenology after the Higgs Discovery*. TH is supported by the DFG Research Training Group GK-1940, *Particle Physics Beyond the Standard Model*. The authors acknowledge support by the state of Baden-Württemberg through bwHPC and the German Research Foundation (DFG) through grant no INST 39/963-1 FUGG (bwForCluster NEMO). This work was supported by the DFG under Germany's Excellence Strategy EXC 2181/1 - 390900948 *The Heidelberg STRUCTURES Excellence Cluster*.

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
