# Peer review of "Two Invertible Networks for the Matrix Element Method"

_SciPost Physics, doi:SciPost Phys. 15, 094 (2023)_

## Round 2 · Referee Report · Sebastien Wertz (Referee 1) · 2022-11-28

Strengths

1- To my knowledge, the first combined use of invertible neural networks (NN) to approximate the so-called transfer functions between parton- and reco-level scattering events appearing in the matrix-element method (MEM), and to numerically evaluate the complex integrals required for the MEM using NN-generated phase-space sampling. 2- Clear explanation of the challenge and of the method used to address it. 3- Relevant example with multiple steps in complexity. 4- Potential for multiple follow-up studies.

Weaknesses

1- It is not clearly demonstrated that the main drawback with traditional MEM implementations, ie their numerical complexity, has been solved.

Report

The authors introduce a novel and interesting approach to the matrix-element method, a high-cost/high-reward method for extracting the maximum information out of a limited dataset at hadron colliders. The computational complexity of the method has always remained a bottlebeck, so that methods shown to be effective in addressing it deserve to be published. The paper is generally well-written. I would therefore like to recommend this submission for publication, but only after the authors have provided more information about the gain in using their new method, compared to more traditional approaches, and addressed my other questions and comments listed below.

1- You mention in the introduction the need to use precision predictions at NLO, but your example only considers LO matrix elements in the calculation of the hard-scattering amplitude. This is different from encoding the effects of QCD radiation in the transfer function, which your method already deals with. Can you elaborate on if and how your method would work with NLO matrix elements?

2- In p.4, you say that you neglect ISR and MPI in the first two samples, then consider ISR with additional hard jets at matrix-element level in the third sample. Can you clarify whether it means that you explicitly disabled ISR in the parton shower for the first two samples, since even without ISR jets at the matrix-element level, Pythia would still be generating some by default?

3- p.6 L4: I do not follow why, in the leptonic case you only consider the parton-level momenta of the top, Higgs and quark, while in the hadronic case all leading-order final-state partons are used (p.14 L-10). Can you explain that choice?

4- p.6 L5: Why not including the missing transverse momentum, which could provide additional information through the (partial) kinematics of the neutrino from the leptonic W decay?

5- In the last step of Eq. (14), you use Bayes' theorem formulated as $p(x_{hard}|x_{reco},\alpha) p(x_{reco}|\alpha) = p(x_{reco}|x_{hard},\alpha) p(x_{hard}|\alpha)$. As you explain, $p(x_{reco}|\alpha)$ is the single-event likelihood at reco-level to be computed, and $p(x_{hard}|\alpha)$ is obtained from the parton-level scattering amplitude, which can be evaluated. Using your cINNs, you can also estimate the conditional probabilities $p(x_{hard}|x_{reco},\alpha)$ and $p(x_{reco}|x_{hard},\alpha)$. This means that, in principle, $p(x_{reco}|\alpha)$ could be computed as $p(x_{reco}|\alpha) = p(x_{reco}|x_{hard},\alpha) p(x_{hard}|\alpha) / p(x_{hard}|x_{reco},\alpha)$ using a single randomly generated configuration $x_{hard}$. Naturally, the cINNs are not perfect, so that making an average over several configurations is likely needed. Still, I would like to see this point clarified in the text. Also, checking that the integrand is indeed independent of $x_{hard}$ could be interesting to judge how well the cINNs perform.

6- You correctly state that the main drawback of the MEM is its computational complexity, but you do not address how much your approach improves on that point. What is the computational cost of training the two cINNs and evaluating the likelihoods of Eq. (16), compared to a brute-force calculation of the MEM likehoods using smart phase-space mapping techniques as implemented in tools such as MadGraph/MadWeight or MoMEMta? You mention in p.12 that you evaluate the integrals using 100k sampling points per event, but that is comparable to what is used in practice to compute "traditional" MEM integrals, and you still need to evaluate the matrix elements in (16), which is the dominant part.

7- It would be interesting to validate the likelihoods obtained using the proposed method against the likelihoods obtained with a traditional implementation of the MEM. But, I will not insist on that point, as the likelihood curves already show that the evaluated likelihoods can be used in measurements.

8- The calibration curves in Figs. 8, 10, 13, provide interesting information in that they compare the parton-level and reco-level measured CP phase for a given sample of events. However, they are not what is usually referred to as a calibration curve. Rather, one would like to see the measured value of the CP phase, averaged over a large number of pseudo-experiments, for different values of the true (fixed) phase, in order to assess the bias and coverage of the method.

9- p.9 L7: Isn't $x_{hard}$ rather sampled from $p(x_{hard}|x_{reco}, \alpha)$?

10- p.16 L5: typo in "is remains efficient"

11- p.17: Do you have an explanation for that under-estimation of $\alpha$?

  • validity: high
  • significance: high
  • originality: high
  • clarity: high
  • formatting: excellent
  • grammar: excellent

Author:  Theo Heimel  on 2023-02-20  [id 3384]

(in reply to Report 1 by Sebastien Wertz on 2022-11-28)

Thank you for your report and suggestions. Please find our replies below. We also attached a file diff.pdf showing our changes in the text in detail.

1- You mention in the introduction the need to use precision predictions at NLO, but your example only considers LO matrix elements in the calculation of the hard-scattering amplitude. This is different from encoding the effects of QCD radiation in the transfer function, which your method already deals with. Can you elaborate on if and how your method would work with NLO matrix elements?

-> We clarified the relation to NLO (which we deliberately skip in this paper) at the end of the introduction. The actual NLO implementation has to wait for a dedicated paper, but first we have to solve the jet combinatorics problem.

2- In p.4, you say that you neglect ISR and MPI in the first two samples, then consider ISR with additional hard jets at matrix-element level in the third sample. Can you clarify whether it means that you explicitly disabled ISR in the parton shower for the first two samples, since even without ISR jets at the matrix-element level, Pythia would still be generating some by default?

-> The ISR jets are generated in Pythia. In the samples without ISR, this is disabled in the Pythia settings and there are no ISR jets. We added a remark in the text.

3- p.6 L4: I do not follow why, in the leptonic case you only consider the parton-level momenta of the top, Higgs and quark, while in the hadronic case all leading-order final-state partons are used (p.14 L-10). Can you explain that choice?

-> There was a typo in the text. The sentence on p.14 was referring to the reco-level phase space, not the hard-scattering phase-space. The latter is the same in all three cases.

4- p.6 L5: Why not including the missing transverse momentum, which could provide additional information through the (partial) kinematics of the neutrino from the leptonic W decay?

-> See the reply to question 4 of the second referee. We clarified that in the text.

5- In the last step of Eq. (14), you use Bayes' theorem formulated as p(xhard|xreco,α)p(xreco|α)=p(xreco|xhard,α)p(xhard|α). As you explain, p(xreco|α) is the single-event likelihood at reco-level to be computed, and p(xhard|α) is obtained from the parton-level scattering amplitude, which can be evaluated. Using your cINNs, you can also estimate the conditional probabilities p(xhard|xreco,α) and p(xreco|xhard,α). This means that, in principle, p(xreco|α) could be computed as p(xreco|α)=p(xreco|xhard,α)p(xhard|α)/p(xhard|xreco,α) using a single randomly generated configuration xhard. Naturally, the cINNs are not perfect, so that making an average over several configurations is likely needed. Still, I would like to see this point clarified in the text. Also, checking that the integrand is indeed independent of xhard could be interesting to judge how well the cINNs perform.

-> In principle, this check is what we are doing when we perform the phase space integration. There are still error bars from the Monte Carlo Integration in Figs. 7, 9 and 12, so it is by far not enough to look at a single sample and there will be a distribution of the integration weights around 1 (as it is the case for every Monte Carlo integration). We added a short remark to make this clearer.

6- You correctly state that the main drawback of the MEM is its computational complexity, but you do not address how much your approach improves on that point. What is the computational cost of training the two cINNs and evaluating the likelihoods of Eq. (16), compared to a brute-force calculation of the MEM likehoods using smart phase-space mapping techniques as implemented in tools such as MadGraph/MadWeight or MoMEMta? You mention in p.12 that you evaluate the integrals using 100k sampling points per event, but that is comparable to what is used in practice to compute "traditional" MEM integrals, and you still need to evaluate the matrix elements in (16), which is the dominant part.

-> The main benefit of our method is that we extract the transfer function from the training data, therefore allowing for a more versatile treatment of QCD and detector effects compared to conventional methods of modeling the transfer functions. As you note, the number of integrand evaluations is comparable to tools like MadWeight/MoMEMta. That means that with our approach of using two cINNs, we were able to use more expressive transfer functions without increasing the numerical complexity. We changed our introduction to clarify this point.

7- It would be interesting to validate the likelihoods obtained using the proposed method against the likelihoods obtained with a traditional implementation of the MEM. But, I will not insist on that point, as the likelihood curves already show that the evaluated likelihoods can be used in measurements.

-> We did not add a comparison to established MEM tools because our method is aimed at situations where an improved modeling of the transfer functions than provided by these tools is necessary. For that reason, we used the measurement of the CP phase as a toy example, but did not include any comparisons to other methods for that specific process.

8- The calibration curves in Figs. 8, 10, 13, provide interesting information in that they compare the parton-level and reco-level measured CP phase for a given sample of events. However, they are not what is usually referred to as a calibration curve. Rather, one would like to see the measured value of the CP phase, averaged over a large number of pseudo-experiments, for different values of the true (fixed) phase, in order to assess the bias and coverage of the method.

-> We agree that this is not what is usually referred to as a calibration curve. However, a scan over a large number of different CP phases would be computationally costly, so we show these approximate calibration curves instead. We added a short explanation in the text.

9- p.9 L7: Isn't xhard rather sampled from p(xhard|xreco,α) ?

-> done

10- p.16 L5: typo in "is remains efficient"

-> done

11- p.17: Do you have an explanation for that under-estimation of α ?

-> We are currently studying the reasons why the performance gets worse after including ISR in more detail and we hope to address this in our next paper on the topic. For now, we do not include an explanation in the text.

Attachment:

diff.pdf

---

## Round 2 · Referee Report · Anonymous (Referee 2) · 2022-11-30

Strengths

1- First use Invertible Networks for the Matrix Element Method, therefore a novel method 2- Inclusion of uncertainty estimations via inclusion of Bayesian Transfer-cINN 3- Study of method applied to a specific, relatively complex use case of CP determination in associated Higgs and single top production

Weaknesses

1- Introduction misses the mark on some key points on the advantages and challenges of the Matrix Element Method and does not acknowledge prior work in this general approach of deep learning for the MEM. 2- Difficult to understand the potential impact given lack of comparison to simpler analysis methods of the chosen physics application example and traditional calculation in terms of computational complexity

Report

1 - The abstract and introduction frame the key benefit of the MEM as analysis of data at colliders in "small event numbers". While it is true that this is a strength of the method (e.g. spin/CP analysis of ZZ->4l, or a small number of signal events generally), this undersells the strengths and broader applicability of the method. As a multivariate analysis approach, the ME method brings in several unique and desirable features, most notably it (1) does not require training data being an ab initio calculation of event probabilities, (2) incorporates all available kinematic information of a hypothesized process, including all correlations, and (3) has a clear physical meaning in terms transition probabilities within the framework of quantum field theory. It might be good to clarify that you mean signal events when you refer to event numbers in the spirit of (2) above, as the technique is often used for analysis of a large number of events in practicle

2 - In the Introduction section (2nd paragraph), you might allude to previous work that has already showed that "modern ML can enhance the MEM". The idea of applying deep learning method to accelerate and sustain MEM calculations goes back to at least 2017 (https://doi.org/10.5281/zenodo.4008241 which is a DOI for https://hepsoftwarefoundation.org/cwp/hsf-cwp-018-CWP_sustainable_matrix_element_method.pdf) and maybe earlier, albeit most of the work is not published in journals (aside from proceedings). You should at least cite "Matrix element regression with deep neural networks — Breaking the CPU barrier" which was published in JHEP_ 04 (2021) 020.

3 - Cuts and Table 1: You mention in the text in reference to Table 1 that you make "detector-level cuts" which i assume from the earlier discussion refers to events after Delphes simulation. Do you have any generator-level cuts that restrict pt or eta of generated particles? It would be good make clear in the text about this point.

4 - On page 6: Why do you not include the neutrino momenta in your 4-momenta list for the leptonic case (Eq 2)? With focus on only the H->yy decay, it seems to me that you could do normal quadratic calculation of p_v_z with W mass constraint and use the MET_x and MET_y for p_v_x and p_v_y, respectively. Is it becuase the resolution on the measured neutrino moment is poor compared with the leptons and jets as to be not useful? Some elaboration in the text would be appropriate in any case.

5 - On page 6, you mention that you disallow ISR radition for simplicity. Could you comment on impact of this simplifying assumption on your study (you do motivate in the subsquent sentense the challenge it poses)? The non-zero (transverse) momentum of the initial state is a challenging complication in MEM analyses as you mention, unless either selections mitigate the effect or one uses an ISR jet reconstruction to boost into the rest frame of the hard scattering system (which has the issue you mention as well).

6 - On page 6: You might also clarify for the reader what you mean that "jets become polluted". I believe i understand what you mean, but it might not be evident to the general reader and it is bit vague and colloquial as written.

7 - On page 7: Not sure why you need to say "hard" in front of cuts. It is either a cut or not. The use of hard twice in the same sentense is also not ideal

8 - On page 7: You cite a website for your lecture notes. You might consider obtaining a persistent identifier (e.g. DOI from Zenodo) for this content and referencing that.

9 - On page 17: You describe that you observe a downward bias in your CP-angle estimation. Do you understand or have any hunch as to the source of this bias with you method?

General: * Have you compared the CP-angle precision you obtain to simpler (non-MEM) method(s) for the same data sets * Have you compared the actual MEM calculation you obtain with MEM codes like MadWeight or MoMemta? * You mention that computational complexity is a major challenge with traditional MEM methods, which is true and has motivated prior work by others on DNN-based surrogate models. Have you compared the MEM evaluations from your code with other codes that do calculations via numerical integration (e.g. MoMemta)?

Requested changes

See above

  • validity: high
  • significance: high
  • originality: top
  • clarity: high
  • formatting: excellent
  • grammar: excellent

Author:  Theo Heimel  on 2023-02-20  [id 3385]

(in reply to Report 2 on 2022-11-30)

Thank you for your report and suggestions. Please find our replies below. We also attached a file diff.pdf showing our changes in the text in detail.

1 - The abstract and introduction frame the key benefit of the MEM as analysis of data at colliders in "small event numbers". While it is true that this is a strength of the method (e.g. spin/CP analysis of ZZ->4l, or a small number of signal events generally), this undersells the strengths and broader applicability of the method. As a multivariate analysis approach, the ME method brings in several unique and desirable features, most notably it (1) does not require training data being an ab initio calculation of event probabilities, (2) incorporates all available kinematic information of a hypothesized process, including all correlations, and (3) has a clear physical meaning in terms transition probabilities within the framework of quantum field theory. It might be good to clarify that you mean signal events when you refer to event numbers in the spirit of (2) above, as the technique is often used for analysis of a large number of events in practicle

-> Thank you for the comment - we expanded the first paragraph to clarify these points.

2 - In the Introduction section (2nd paragraph), you might allude to previous work that has already showed that "modern ML can enhance the MEM". The idea of applying deep learning method to accelerate and sustain MEM calculations goes back to at least 2017 (https://doi.org/10.5281/zenodo.4008241 which is a DOI for https://hepsoftwarefoundation.org/cwp/hsf-cwp-018-CWP_sustainable_matrix_element_method.pdf) and maybe earlier, albeit most of the work is not published in journals (aside from proceedings). You should at least cite "Matrix element regression with deep neural networks — Breaking the CPU barrier" which was published in JHEP_ 04 (2021) 020.

-> The first paper seems to not be on the arXiv and not accessible through Inspire, so we do not really see the point of citing such an internal report. We did add the second reference.

3 - Cuts and Table 1: You mention in the text in reference to Table 1 that you make "detector-level cuts" which i assume from the earlier discussion refers to events after Delphes simulation. Do you have any generator-level cuts that restrict pt or eta of generated particles? It would be good make clear in the text about this point.

-> We only apply cuts at the detector level. We clarified that in the text.

4 - On page 6: Why do you not include the neutrino momenta in your 4-momenta list for the leptonic case (Eq 2)? With focus on only the H->yy decay, it seems to me that you could do normal quadratic calculation of p_v_z with W mass constraint and use the MET_x and MET_y for p_v_x and p_v_y, respectively. Is it becuase the resolution on the measured neutrino moment is poor compared with the leptons and jets as to be not useful? Some elaboration in the text would be appropriate in any case.

-> While this could be done, a reconstruction of the neutrino momentum as a function of the other momenta would keep the number of degrees of freedom the same. Especially in the case of the Transfer-cINN, that might make the training less stable since it would restrict the training data to a sub-manifold of the larger input space. We added a remark in the text.

5 - On page 6, you mention that you disallow ISR radition for simplicity. Could you comment on impact of this simplifying assumption on your study (you do motivate in the subsquent sentense the challenge it poses)? The non-zero (transverse) momentum of the initial state is a challenging complication in MEM analyses as you mention, unless either selections mitigate the effect or one uses an ISR jet reconstruction to boost into the rest frame of the hard scattering system (which has the issue you mention as well).

-> When we generate the training data, we extract the hard scattering momenta before ISR is added. That means that in principle, our networks should be able to perform the boost into the hard-scattering frame without doing so manually. However, this requires the networks to identify which jets are from ISR, so the challenging aspect is again jet combinatorics. We added a short explanation in the text.

6 - On page 6: You might also clarify for the reader what you mean that "jets become polluted". I believe i understand what you mean, but it might not be evident to the general reader and it is bit vague and colloquial as written.

-> We reformulated the sentence.

7 - On page 7: Not sure why you need to say "hard" in front of cuts. It is either a cut or not. The use of hard twice in the same sentense is also not ideal

-> When cuts are applied at the detector-level, this will not result in a hard cut-off at the hard-scattering level. Instead, it will be smeared out. The mentioned assumption is that we can still account for the cuts by replacing the full volume with the fiducial volume. We reformulated that part of the text to make this clearer.

8 - On page 7: You cite a website for your lecture notes. You might consider obtaining a persistent identifier (e.g. DOI from Zenodo) for this content and referencing that.

-> The lecture notes are on arXiv now. We updated the reference.

9 - On page 17: You describe that you observe a downward bias in your CP-angle estimation. Do you understand or have any hunch as to the source of this bias with you method?

-> We are currently studying the reasons why the performance gets worse after including ISR in more detail and we hope to address this in our next paper on the topic. For now, we do not include an explanation in the text.

General: * Have you compared the CP-angle precision you obtain to simpler (non-MEM) method(s) for the same data sets * Have you compared the actual MEM calculation you obtain with MEM codes like MadWeight or MoMemta?

-> The goal of our work was to show that it is possible to use cINNs to learn shower and detector effects in the MEM. We used the measurement of the CP phase as a toy example for that (as we state in the introduction). Therefore, we did not compare the performance to other MEM or non-MEM methods or tools for this specific physics application.

  • You mention that computational complexity is a major challenge with traditional MEM methods, which is true and has motivated prior work by others on DNN-based surrogate models. Have you compared the MEM evaluations from your code with other codes that do calculations via numerical integration (e.g. MoMemta)?

-> The main benefit of our method is that we extract the transfer function from the training data, therefore allowing for a more versatile treatment of QCD and detector effects compared to conventional methods of modeling the transfer functions. Using our approach with two cINNs, we were able to do this without impairing the numerical efficiency. We changed our introduction to make this clearer. MadWeight or MoMemta work by making simplifying assumptions about the transfer function, allowing them to construct corresponding phase space mappings. Since this is not the case for our method, we do not include a comparison with established MEM tools.

Attachment:

diff_4PxnMyE.pdf

---

## Round 3 · Referee Report · Sebastien Wertz (Referee 1) · 2023-3-17

Report
Note: this review is based on the "v4" posted on Arxiv.
I thank the authors for answering my questions and implementing my suggestions (and providing a handy "diff"). I find that the new version improves in clarity. I have a few small remaining comments. Once they are answered, I will be happy to recommend this paper to be accepted for publication.
Requested changes
1- Now that the introduction puts the focus on the modelling of the transfer function, which the method is indeed mainly about, I think the abstract could be rephrased to reflect that: E.g. remove the "but computationally expensive", and explain that in practice we have had to make many assumptions about the form of the transfer function for QCD and detector effects, a limitation which your method lifts without increasing significantly the computational complexity.
2- Typo on p.3 ("to not (yet)")
3- About the statement added regarding the use of the missing transverse momentum: in that particular case (no ISR) you are right, but I just want to point out that in general the MET can provide additional information, e.g. to limit the jet combinatorics if you had included ISR in the leptonic channel.
4- Regarding the clarification added about Eq. (14), I would suggest to rephrase as "our xhard-integration would become trivial if we could sample from..." to make clear that the integration only becomes trivial if that sampling is exact, which won't be the case in practice.
5- Shouldn't the schematic in Figure 4 also be updated with the extra 1/p(r) factor added in the v4?
Anonymous on 2023-03-30 [id 3525]
I am satisfied with the authors responses to my questions and recommend publication after a minor update described below is implemented.
I also note that the authors found a mistake in Eq 16 in the paper which was also included in the code upon which the paper results were based. This correction is fully implemented in v4 of the manuscript and the authors inform that the changes were small so there was no need to make any changes to the discussion of the results.
I thank the authors for significant updates to the introduction section after the recommendations. It reads much better. However, there one small update i would like to see implemented before the article is published. The use of deep learning to address the computational challenges of the MEM calculation was first described (see page 20 in the section on "Sustainable Matrix Element Method" at https://inspirehep.net/files/ca2037001073f35f718b719fa3bddf8c) in "A Roadmap for HEP Software and Computing R&D for the 2020s" which was published in Comput.Softw.Big Sci. 3 (2019) 1, 7. This should be referenced in your introduction in an update before publication.
I would suggest (you should decide on the exact wording for resubmission)
"The connection between the MEM and modern ML-methods was pioneered in Ref. [67]..."
be changed to something like:
"The connection between the MEM and modern ML-methods was first described in Ref.[67] which proposed using deep neural networks to overcome the computation challenges inherent to the MEM. In Ref. [68], the use of a deep neural network built by regression of the MEM integral was first demonstrated."
where
[67] HEP Software Foundation Collaboration; Johannes Albrecht(Dortmund U.) et al., "A Roadmap for HEP Software and Computing R&D for the 2020s; ", Comput.Softw.Big Sci. 3 (2019) 1, 7 , arXiv:1712.06982 [hep-ex]
[68] F. Bury and C. Delaere, Matrix element regression with deep neural networks — Breaking the CPU barrier, JHEP 04 (2021) 020, arXiv:2008.10949 [hep-ex].
It is the opinion of this referee that with this change that your paper would accurately describe the when and where the key ideas of deep learning for MEM were first published. Your novel ideas and approaches described in your manuscript then build (significantly, in my opinion) upon these publications.

---

## Round 3 · List of Changes

We would like to thank the referees for their comments and provide an updated version addressing their requests. A detailed list of changes can be found in our replies to the referees.

---

## Round 5 · List of Changes

We would like to thank the referees for their comments. We uploaded an updated version with the requested changes.

Report by Sebastien Wertz:

  1. We changed the first two sentences of the abstract to "The matrix element method is widely considered the ultimate LHC inference tool for small event numbers. We show how a combination of two conditional generative neural networks encodes the QCD radiation and detector effects without any simplifying assumptions, while keeping the computation of likelihoods for individual events numerically efficient."

  2. We corrected the typo.

  3. Indeed, it might be useful when ISR is included in the leptonic channel, for example to apply a sorting of the reco-level jets that makes the combinatorics easier to extract for the network.

  4. We changed the text to "our xhard-integration would become trivial if we could sample".

  5. We updated Figure 4 to include 1/p(r).

Anonymous report:

We changed the part of the introduction about earlier uses of ML for the MEM to "A connection between the MEM and modern ML-methods, mentioned in Ref. [67], was demonstrated in Ref. [68], specifically using a deep regression network to evaluate the MEM integral."

[67] HEP Software Foundation, J. Albrecht et al., A Roadmap for HEP Software and Computing R&D for the 2020s [68] F. Bury and C. Delaere, Matrix element regression with deep neural networks — Breaking the CPU barrier

---

## Editorial Decision

published